# Evaluating U-Space for UAM in Dense Controlled Airspace

**Michal Černý *** , **Adam Kleczatský** , **Tomáš Tlučhoř** , **Milan Lánský and Jakub Kraus**

Department of Air Transport, Faculty of Transportation Sciences, Czech Technical University in Prague, 11000 Prague, Czech Republic; kleczada@fd.cvut.cz (A.K.); tluchto2@fd.cvut.cz (T.T.); lanskmi1@fd.cvut.cz (M.L.); jakub.kraus@cvut.cz (J.K.)

\* Correspondence: cernym52@fd.cvut.cz

**Abstract:** The operation of unmanned aircraft systems in shared airspace can serve as an accelerator for the global economy and a sensitive addition to the existing mix of transportation modes. For these reasons, concepts of Unmanned Traffic Management have been recently published, defining advanced rules for all potential participants in the operation of unmanned systems. Airspace primarily dedicated to automated unmanned system operations, referred to as U-space in Europe, needs to be designated with consideration for the surrounding airspace. This is especially important in cases where the airspace is controlled, and when declaring U-space airspace, it is necessary to pay particular attention to the density of surrounding air traffic. The goal of this article is to assess the suitability of establishing U-space airspace for Urban Air Mobility in terms of traffic density in a controlled area above the selected metropolis, which is Prague, Czech Republic. To achieve this goal, data on air traffic in the given area were analyzed to obtain precise information about the traffic distribution. Areas in which the establishment of U-space airspace is possible both without implementing dynamic reconfiguration and with the application of the dynamic reconfiguration concept were also selected. The result is the determination of whether it is possible to establish U-space in airspace, as in the analyzed case of the Ruzyně CTR, U-space can be introduced in 83 % of the territory.

**Keywords:** U-space; CTR; UAM; UAS; UTM; controlled airspace

## 1. Introduction

Unmanned aircraft systems (UASs) are no longer just the domain of amateur pilots. The rapid development they have undergone in the last decade has enabled their use in industries such as agriculture, transportation, and parcel delivery to end users. However, for its more efficient utilization, it is essential to operate them beyond the visual line of sight (BVLOS). To create a safe and efficient environment, the concept of UTM (Unmanned Traffic Management), known as U-space in Europe, has been developed. In this airspace, it will be possible to operate UASs in areas BVLOS by providing services that will support operations and make them more efficient and safe. It is expected that the volume of UAS operations will increase exponentially [1]. Currently, a highly discussed application involves Urban Air Mobility (UAM), which pertains to the transport of people and cargo within urban areas.

The use of UASs for passenger transport within large cities, where helicopters have been and are still used, is significantly more cost-effective, environmentally friendly, and overall more accessible than similar helicopter transportation [2]. Another benefit is the expected reduction in noise pollution and emissions. Due to the multiengine design, UAS operations are also much safer, which is a significant advantage, especially in urban areas. In relation to urban areas, different problems arise in the form of privacy concerns of citizens [3]. In reaction, some municipalities passed legislation preventing UAS from operating close to homes or above private property.

According to the study conducted in [4], passengers traveling to the airport are highly sensitive to price. Passenger transport using UASs is expected to be a more cost-effective option compared to the helicopters currently used. However, passengers can also choose

other modes of transportation, such as buses, trains, or personal cars. The study suggests that mainly older passengers are willing to pay extra for privacy, i.e., private UAS transport. However, it is suggested that even under the best scenario, the port infrastructure will not be within walking distance for most people [5].

Currently, due to the plans regarding UTM in the EU, it is crucial to put all operations that would be conducted without a pilot on board into U-space airspace.

U-space is an airspace designated for the operation of UASs. Commission Implementing Regulation (EU) 2021/664 [6] defines U-space airspace as a "geographical zone for UAS defined by the Member States, where the operation of UAS is allowed only with the support of U-space services." These U-space services, based heavily on automation, are designed to facilitate safe and efficient access to U-space. Harmonized EU legislation also distinguishes other geographical zones that are not U-space and may be declared for the purpose of safeguarding the health of persons, property, or nature. Although Member States may establish conditions for UAS operations in these zones, they are not defined as U-space and do not have dedicated services. Regarding U-space airspace established up to 150 m Above Ground Level (AGL) as recommended by AMC/GM [7], it is essential to introduce a tool called Dynamic Airspace Reconfiguration (DAR). DAR enables Air Traffic Service Providers (ATSPs) in specific airspace to dynamically reconfigure U-space airspace to accommodate short-term changes in demand for crewed aircraft operations.

The operations in U-space can involve UASs of various designs and sizes. U-space serves as an accelerator for the further advanced development of UAS operations, which may carry not only cameras, but also, for example, passengers or (dangerous) goods. The general concept of personal and cargo transport using UASs, including missions in urban, regional, and interregional areas, is called UAM or Advanced Air Mobility (AAM) [8]. One of the main concepts and studies intended [9,10] for the utilization of UAM is the transport of passengers to/from airports. Research also addresses possibilities for traffic organization in these areas (corridors) and the management of heliport occupancy. The results suggest that UAS operations clustered closer together in corridors produce less visual pollution than free flight [11].

Initially, UAM flight routes will be established in areas where there will be no conflict with the arrival and departure routes from the respective airports. Therefore, they will not be located at altitudes or in proximity to points used for Instrument Flight Rules (IFRs) routes. Potential conflicts between UAM and flight routes would impose a higher workload on air traffic controllers [12], which should be avoided. Air traffic control with a high workload could create a hazardous operational environment near airports within controlled airspace.

Controlled airspace is a portion of airspace where air traffic control (ATC) services are provided. It is further divided into classes based on vertical divisions of the airspace, which determine the rules for entering and operating within that airspace. In areas around controlled airports, Control Zones (CTRs) are established to protect departing and arriving aircraft. These areas also experience relatively dense air traffic and are subject to control by the responsible Air Navigation Service Provider (ANSP). Above and around the CTR, there is a Terminal Control Area (TMA), or TCA in the U.S. and Canada, with a lower limit of at least 700 feet. The requirements for UAS operations under UTM are expected to include scenarios that characterize specific transport of people and cargo within urban areas and metropolitan regions. There is an inherent overlap of interest here, as most metropolitan areas are in close proximity to one or more airports, which pose increased demands for defining controlled airspace in their vicinity.

The study in [13] focusing on the identification and prediction of UAM availability for the Sao Paulo airport examined departing and arriving aircraft within a distance of 40 nautical miles from the center of the airport. The goal was to determine arrival and departure routes for UAM with the fewest conflicts with manned aircraft. The analysis utilized data from FlightRadar24. However, it only took into account operations that began or ended at the airport, excluding flights that merely transited through the area. The study's

outcome involved identifying arrival and departure clusters, based on which areas with lower traffic intensity were identified, along with their relevance to UAM operations near the Sao Paulo airport.

In accordance with the possibility of establishing U-space airspace, it is necessary to address the method of assessing the safety of planned operations conducted in it. In AMC/GM to Regulation 2021/664 (U-space), it is stated that the determination of U-space airspace is influenced by safety, privacy, or environmental factors. Among significant safety factors, factors related to the "type, density, and complexity of current and planned manned operations" are mentioned, among others. These are the principles for assessing the potential creation of U-space airspace, considering the level of impact on the existing air traffic structure. AMC/GM provides a "Checklist template" that classifies risks on the ground and in the air. In total, approximately 37 assessment parameters are analyzed using this template. However, these parameters do not have a quantification scale in the legislative proposal—in other words, I know what I am evaluating, but not how. Among the selected risks addressed by this article are the layout of ATS routes, the location of the CTR and TMA, and operational restrictions (TRA, TSA, TMZ, etc. or airport operating hours). Arrival and departure routes or transit routes are assessed in the case of IFR operations and in the case of VFR operations, for example, common VFR routes and corridors, as well as low-altitude military operations.

The current legislation [14] anticipates that UAM operations will be conducted in the specific or certified category. Operations related to passenger or dangerous goods transport will always be part of the certified category. Operational requirements include the need to ensure safety, for example, by lowering ground risk. The ground risk includes all scenarios that could lead to a UAS crash and therefore a fall to the ground including accidents resulting from UAS malfunction, midair collision between UASs, or collisions with manned aircraft. The ground risk does not differentiate between different causes of the fall. Damage to the ground is assessed.

In addition to ground risk, air risk is also assessed. Currently, it is primarily evaluated based on the traffic density in a given area. One of the most high-risk areas is in the vicinity of controlled airports, where traffic density is high, and, most notably, there are operations of large transport aircraft, where the potential consequences of an airborne collision are enormous. In the past, studies have been conducted to model the probability of an air collision between two manned aircraft [15]. However, determining the probability of a collision between a manned aircraft and a UAS is more complex using these methodologies. UAS operations currently do not take place within predefined flight paths to which the methodology could be applied.

EUROCONTROL has also published the methodology "U-space Airspace Risk Assessment" [16], which aims to describe a methodology for conducting an Air Risk Assessment (ARA) in support of the designation of U-space airspace(s) by states. The document addresses risks associated with both people on board an aircraft in flight and those on the ground. The methodology categorizes data collection as one of the initial phases of ARA, which includes, among other things, aeronautical data. These data encompass information about air traffic, describing its characteristics and volume.

The genesis of U-space airspace is described through a systematic assessment process; see Figure 1. This assessment is characterized by a proactive intent to initiate the designation of U-space airspace, primarily driven by considerations related to safety, security, environmental concerns, or privacy. In the initial preparation phase, the scope is clearly defined. This involves establishing the context and purpose of the U-space operation and addressing key questions related to the motivation behind the creation of this airspace. Subsequently, the reference scenario phase ensues, during which a comprehensive analysis of the operational landscape is conducted, preceding any formal declaration of U-space airspace. Data essential for supporting the decision-making process are meticulously collected during this stage. It should be noted that at this point, the stakeholders with whom interviews are conducted have already shown a vested interest. The assessment phase

follows, yielding results in the form of stipulated requirements for UASs and services to be provided within this airspace. In addition, a safety, security, privacy, and environmental risk assessment is performed. This phase serves as a validation process. Upon its successful completion, the formal designation of U-space airspace becomes feasible.

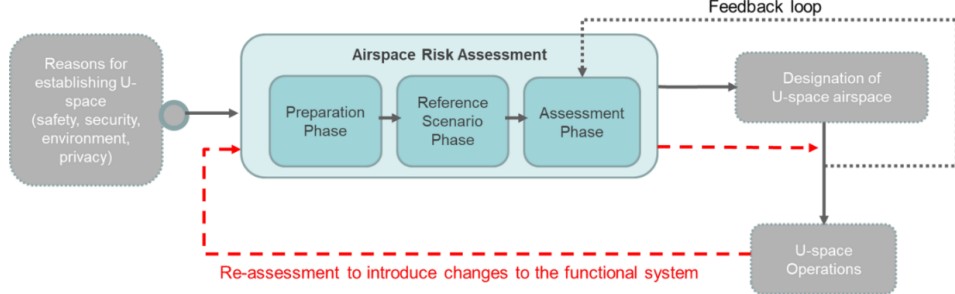

**Figure 1.** The phases of a U-space airspace risk assessment [16].

Assessing the impact of establishing U-space airspace is also crucial due to DAR, which should be used as minimally as possible within the established U-space airspace [7]. Therefore, there is a logical need for knowledge about the utilization of the specific volume of airspace by manned aircraft before defining U-space airspace in a given location. In cases where the significance of the impact on manned aircraft is not assessed, it can often lead to the activation of DAR with negative implications for the safety and efficiency of unmanned operations in U-space. The need for integration is based on the fundamental principles of sophisticated prevention.

## 2. Materials and Methods

The potential impact of the introduction of U-space can be assessed by analyzing archived data that interpret individual flights. These data can come from both primary surveillance radar (PSR) and secondary surveillance radar (SSR). Aviation is based on precise and standardized measures to ensure safe and efficient aircraft operations. Height, altitude, and flight level are fundamental concepts that are used to define vertical positioning in the airspace. Additionally, the implementation of advanced technologies, such as Automatic Dependent Surveillance-Broadcast (ADS-B), has revolutionized the way aircraft transmit and receive data for surveillance purposes.

### 2.1. Data Sources

The methodology involves the data analysis of secondary radar transponder information and its depiction at various flight heights up to 500 ft and 1000 ft AGL. The observed area, encompassing the Ruzyně CTR, has dimensions of 38 × 28 km and extends to the Vodochody CTR and Kbely CTR, as shown in Figure 2. Based on the quantification of air traffic, areas with lower traffic intensity are subsequently identified, which could potentially be candidates for the establishment of U-space airspace.

The analysis is based on data from secondary radar transponders in ASTERIX (All-Purpose Structured Eurocontrol Surveillance Information Exchange) format, which was provided by the Air Navigation Services of the Czech Republic. ASTERIX CAT062 is a specific data category within the ASTERIX data format used in aviation [17]. The collected data described the air traffic for each day in the month of June 2022, with traffic continuously recorded throughout the 24 h period. ASTERIX CAT062 is designed to transmit aircraft-derived data, including critical information such as the aircraft's position, velocity, and identification data. It provides a standardized and efficient means of exchanging surveillance information between aircraft and ground systems. The ASTERIX CAT062 data format follows a structured format, with different subfields dedicated to specific data elements [18], including the three-dimensional position of the aircraft, velocity vectors, identification parameters, and other relevant information essential for air traffic control and surveillance purposes.

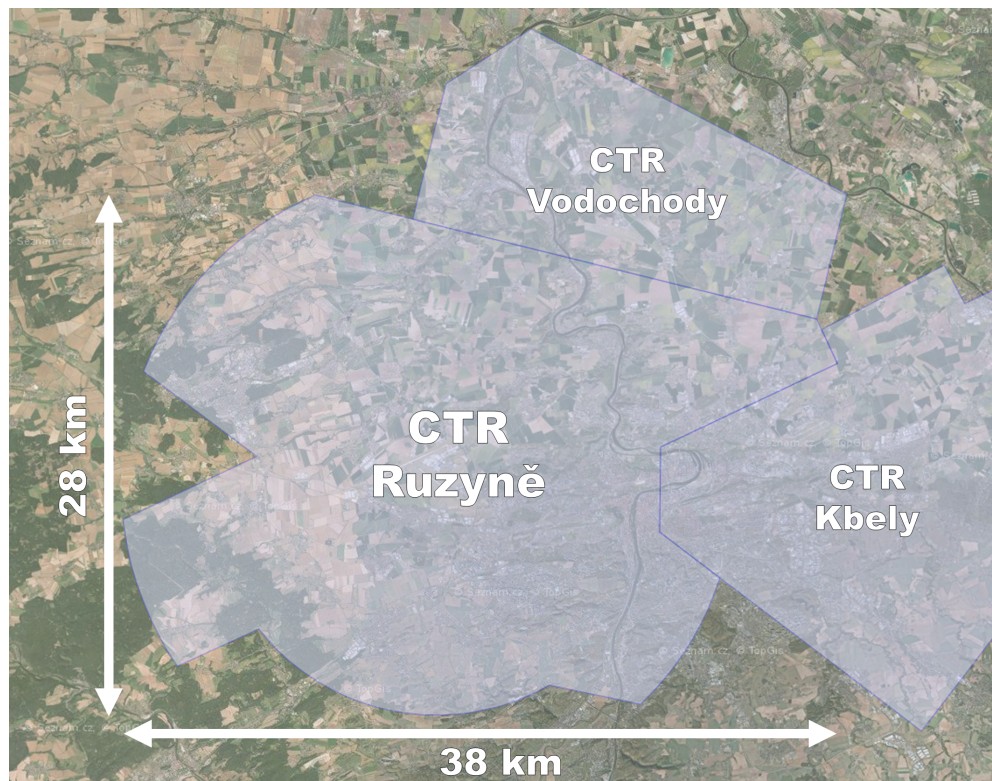

**Figure 2.** The monitored area, including the Ruzyně CTR, has dimensions of 38 × 28 km.

The processed data from the ASTERIX system describe time of reception of the transponder message with precision to one-thousandth of a second (IO62/070), aircraft position expressed in latitude and longitude using the WGS-84 format (IO62/100), aircraft altitude in feet (IO62/136), and aircraft call sign (IO62/245); see Table 1. It should be noted that the altitude of the aircraft was measured barometrically and, therefore, referenced to the pressure of 1013.25 hPa at sea level according to the International Standard Atmosphere.

**Table 1.** Used ASTERIX data table description.

| Data Code | Description of the Data |
| --- | --- |
| IO62/070 | Time of reception of the transponder message with the precision of one-thousandth of a second |
| IO62/100 | Aircraft position expressed in latitude and longitude using the WGS-84 format |
| IO62/136 | Aircraft altitude in feet |
| IO62/245 | Aircraft call sign |

In total, there were 565,152 records of aircraft positions corresponding to the time period from 01:03 on 1 June to 23:59 on 30 June 2022. Records of secondary radar responses that did not contain aircraft call signs were excluded from the data set. Approximately 3% of the total number of records, which was 17,039 records, were removed for this reason. This step was taken because it would not have been possible to determine whether these records represented a single flight or multiple consecutive flights. Without call sign information, it would have been challenging to track individual flight trajectories accurately and there could have been confusion between the flight paths of multiple unidentified flights.

### 2.2. Altitude Correction

The ANSP provided additional pressure data from each hour of the days of June 2022 related to specific locations, which allowed for the accurate calculation of the aircraft's altitude. To convert the pressure deviation into altitude, a table used by the ANSP was

utilized. This table specifies the altitude correction in feet for each pressure value. This method is more precise than the general rule, which states that every 1 hPa difference between the current pressure and the International Standard Atmosphere (ISA) results in a 27 ft altitude difference. As shown in Figure 3, this method follows a part of a parabolic curve rather than a linear simplification, as suggested by the commonly accepted rule. Therefore, it provides a more accurate and realistic representation.

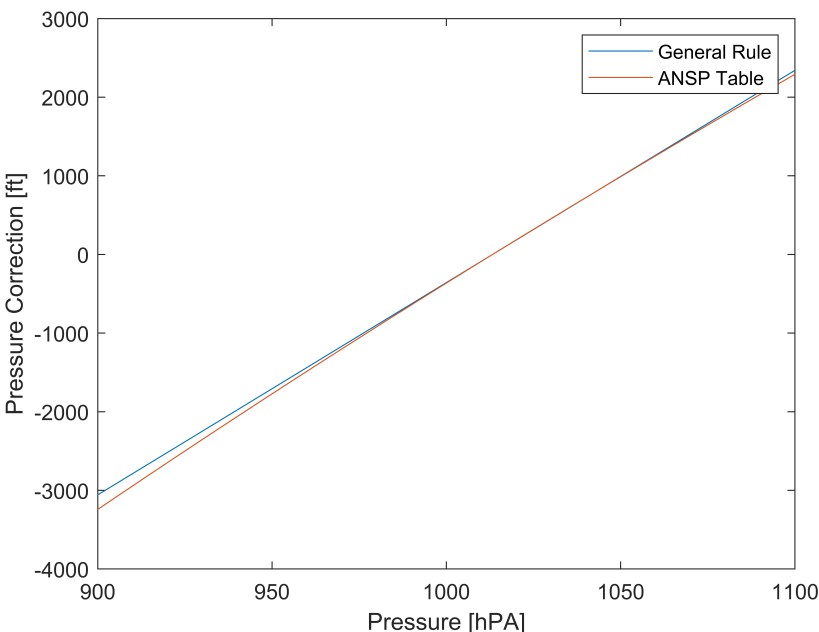

**Figure 3.** Difference between ANSP pressure table and general rule.

In the next step, the altitudes above sea level were converted into values that represent the height above the terrain. A Digital Elevation Model (DEM) from Copernicus [19] was used as the reference data set. This model was chosen because it is publicly accessible and provides sufficient accuracy and density of elevation values. The digital elevation model offers a raster with elevation values at a resolution of 30 m. The recorded altitude of the aircraft was always corrected based on the nearest DEM value. From the adjusted data, records below 500 ft and below 1000 ft above terrain were subsequently selected, as shown in Figure 4.

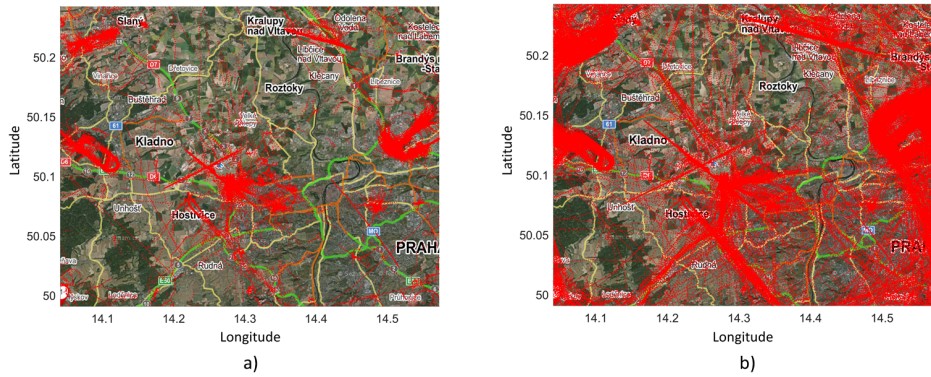

**Figure 4.** Visualisation of ANSP data: (**a**) below 500 ft, (**b**) below 1000 ft.

### 2.3. Traffic Density Analysis

During UAS operations, the common 1:1 rule is applied, which means that on both sides of the flight path, there must be a buffer zone at least as wide as the height of the flight. This rule helps ensure that the UAS remains within the designated corridor. Current

regulations allow UAS flights up to a height of 120 m. To illustrate UAS operations in the Ruzyně CTR, a grid method was used. The CTR was divided into a square grid network, with each grid cell size determined by the 1:1 rule, at 250 × 250 m. This translates to 120 m on each side of the UAS flight path, with an additional bonus buffer. Subsequently, it was determined how many individual flights occurred within each of these grids up to a height of 1000 feet above the terrain. The flight counts were based on the call signs assigned in the ASTERIX messages for each record. Since helicopters from the Czech Police Aviation Service and the air rescue service also operate within the CTR, a time interval of 300 s was used as the threshold to distinguish between two separate flights. This means that if an aircraft passed through a grid with a gap of more than 5 min, this transit was counted as a separate and distinct flight.

As part of the methodology, it was decided that a grid with low traffic density would be considered where there are flights at an average quantity of less than 1 flight per week, meaning fewer than 4 flights per month. In such cases, this means that those are not regular flights. A maximum of 1 airspace reconfiguration per week is an acceptable value. The value of less than 1 flight per week comes from GM1 to Article 4 of U-space regulation [20], where it is stated that "the number of instances where DAR would be required should be limited". It is important to note that in this decision-making process, no distinction is made when assessing flights from air rescue services, security forces, or general aviation (GA). The result of the analysis is a value that describes the percentage of grids that meet the specified conditions, i.e., have fewer than 4 flights in the observed month. This allows one to determine the percentage of the chosen area in the Ruzyně CTR that can accommodate the creation of U-space.

Within the defined area, a different number of call signs for aircrafts were detected each day throughout the month, as shown in Figure 5. The lowest number below 1000 ft AGL was observed on 4 June, with a total of 281 unique call signs detected. The highest number of call signs below 1000 ft AGL was observed on 17 June, with a count of 437. Below 500 ft AGL, the lowest number was 279 unique call signs detected on 4 June, and the highest was 435 on 17 June.

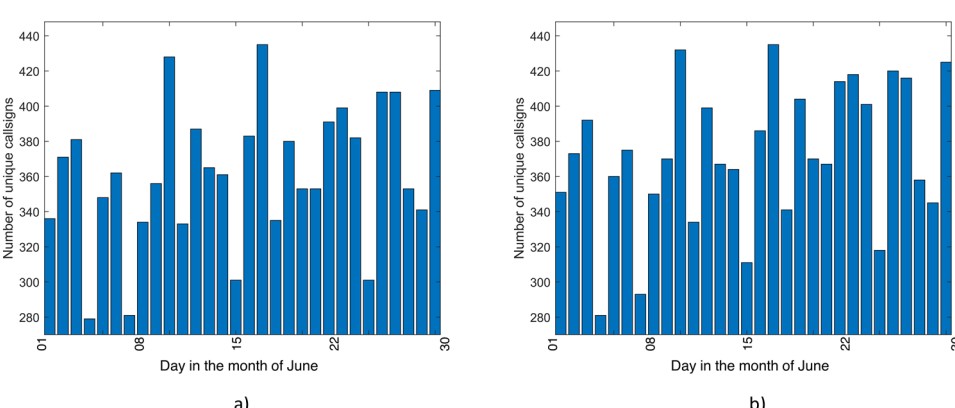

a)  b)

**Figure 5.** Number of unique call signs daily in the defined area: (**a**) below 500 ft AGL, (**b**) below 1000 ft AGL.

This variability indicates the diversity and dynamics of air traffic in this area. This variability is likely attributed to the varying levels of activity at Václav Havel Airport throughout the week. Additionally, the number of recreational flights in general aviation may also change depending on weekdays or weekends. Variable weather conditions also play a role in this dynamic environment.

Using unique call signs, the main users of airspace have been identified. The list includes commercial airlines operating from Vaclav Havel Airport, flights of security forces, helicopter emergency medical service (HEMS), and general aviation (including VFR flights) operating from smaller airports in the vicinity of Prague such as Tocna, Vodochody, Letnany,

and Kbely. Regarding Kbely and Vodochody, some flights are of a military nature. Even though these air traffic data are from June 2022, their use for the traffic evaluation for the future UAM integration is appropriate due to the inclusion of all air operation types that are active in defined areas, while also including their trajectories. The amount of traffic near Vaclav Havel Airport Prague clearly shows the approach and departure routes used by air transport aircraft. The area including those routes is unsuitable for UAM operations due to the high volume of traffic, which is more than 1 flight per week as the methodology establishes, which makes it unsuitable for U-space airspace creation. The biggest uncertainty lies in the operations of security forces and HEMS because of their ad hoc flight routes depending on the current mission. However, in the current roadmap of these flight services, there is no expected increase in the volume of operations. So, the volume of operations from 2022 can be considered to be the same in the following years.

## 3. Results

Analysis of secondary radar transponder data, after applying the methodology outlined in the previous section, yielded valuable information on the traffic distribution within the Ruzyně area CTR. Examination of corrected data revealed significant traffic flows within the Ruzyně area CTR; see Figure 6. The spatial patterns of the movements of the aircraft provided a comprehensive understanding of the distribution of traffic in the airspace. The following paragraphs present the findings in terms of traffic density, flight paths, and spatial relationships. The density maps generated from the data provided a visual representation of areas with high and low concentrations of aircraft.

The results indicate that the highest traffic density was observed in proximity to major airports and navigational waypoints. These areas experienced a substantial influx of aircraft, resulting in congested airspaces. On the contrary, lower traffic densities were observed in remote or less populated regions within the Ruzyně area CTR. This is also influenced by the existence of restricted airspace R9 (PRAHA), which extends over the city center of Prague.

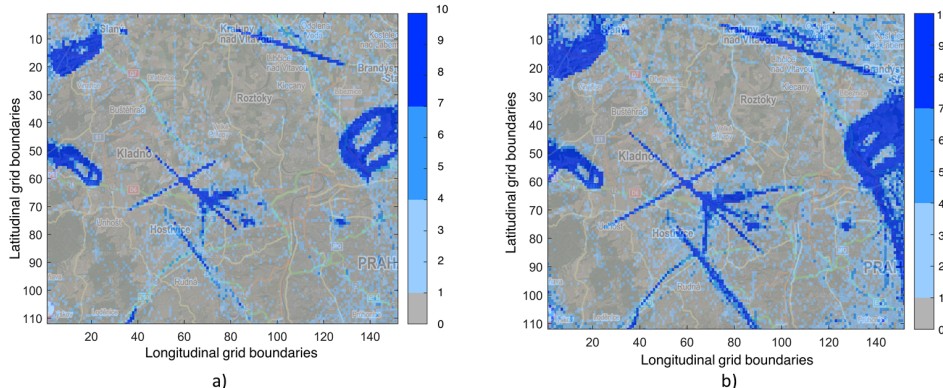

**Figure 6.** Grids representing a number of unique flights expressed by different shades of blue color: (**a**) below 500 ft AGL, (**b**) below 1000 ft AGL.

To introduce an initial U-space airspace, it is essential to avoid areas where conflicts with manned aircraft would be frequent. According to the methodology, areas with low traffic density are defined as grids where an average of three or fewer flights occur per month. Figure 7 shows grids where three or fewer unique flights occurred up to 500 ft and 1000 ft AGL during the month of June 2022. These are represented in white. Grids with a higher number of unique flights, four or more, which do not qualify as suitable areas for U-space creation due to their frequent and dense traffic, are shown in black.

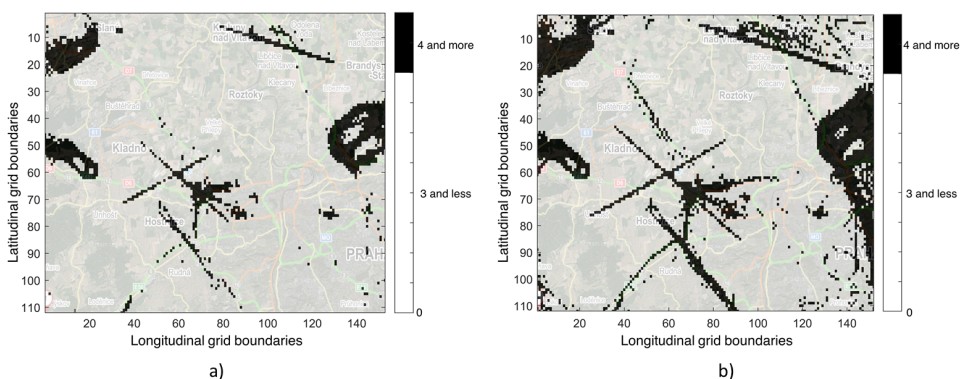

**Figure 7.** Visualization of grids with acceptable traffic density: (**a**) below 500 ft AGL, (**b**) below 1000 ft AGL.

The total area of the observed area measures $38 \times 28$ km, resulting in a total area of 1064 km$^2$. Table 2 illustrates the number of individual grids based on the number of unique flights that occurred within them. It also calculates the areas consisting of different groups of grids based on the number of flights recorded in them. Grids with no, one, two, three, and four or more flights in the observed month were quantified.

**Table 2.** Number of grids with specific number of unique flights with absolute and relative area below 500 ft AGL and below 1000 ft AGL.

| Number of Flights in the Grid | Number of Grids below 500 ft | Number of Grids below 1000 ft | Total Area below 500 ft | Total Area below 1000 ft | Total Area below 500 ft | Total Area Percentage below 1000 ft |
|---|---|---|---|---|---|---|
| 0 | 12,150 | 9324 | 759 km$^2$ | 583 km$^2$ | 71.4% | 54.7% |
| 1 | 2160 | 2567 | 135 km$^2$ | 160 km$^2$ | 12.7% | 15% |
| 2 | 934 | 1464 | 58 km$^2$ | 92 km$^2$ | 5.5% | 8.6% |
| 3 | 415 | 807 | 26 km$^2$ | 50 km$^2$ | 2.4% | 4.7% |
| 4 and more | 1365 | 2862 | 85 km$^2$ | 179 km$^2$ | 8% | 16.8% |

The total area in which no traffic was detected up to 500 ft AGL using the set parameters is approximately 759 km², which accounts for 71.4 % of the total. Grids with three or fewer detected flights have a total area of 978 km². The area with four or more flights was determined to be 85 km².

The total area in which no traffic up to 1000 ft AGL was detected using the set parameters is approximately 583 km² out of the total 1064 km². The area in which, according to the established methodology, it would be possible to establish a U-space, i.e., grids with three or fewer flights, was determined to be 885 km² out of the total 1064 km². The area with four or more flights, thus not meeting the specified conditions for creating a U-space, was determined to be 179 km².

The area with three or fewer flights for both 500 ft and 1000 ft AGL predominantly covers the Ruzyně area CTR. These areas are also contiguous, meaning that suitable grids for U-space implementation are located in continuous regions adjacent to each other. This does not imply that the overall suitable area consists of a large number of small isolated parts that are not suitable for commercial UAS operations in the Ruzyně CTR.

## 4. Discussion

The analysis of flight paths sheds light on the routes taken by aircraft within the Ruzyně area CTR. By examining the sequential data points of individual flights, it was possible to discern common trajectories and identify any deviations or variations. The findings provided information on the preferred routes, approach paths, and departure procedures. Initial observations suggest that most flights followed established airways and instrument flight procedures, especially during the approach phases. These predefined routes ensured

efficient and safe navigation for aircraft operating within controlled airspace. However, the analysis also revealed instances of deviations from standard flight paths, which may have been influenced by factors such as air traffic control instructions, weather conditions, or specific operational requirements. Apart from the standard flight routes, the movement was primarily associated with flights of police helicopters and the air rescue service. This is mainly due to the geographical location of the major Prague hospitals, which are relatively close to the city center and therefore fall within the restricted area R9.

By examining the altitudes recorded in the data set and considering the altitude corrections based on the nearest values from the DEM, valuable information about flight levels and altitude preferences were obtained. The results reveal a diverse range of altitudes observed between aircraft operating within the Ruzyně area CTR. Generally, scheduled air transport flights maintained higher altitudes, outside of the airport vicinity, adhering to the established routes specified by the air traffic control. On the contrary, during the approach and departure phases, the aircraft gradually descended or ascended, respectively, as they navigated the assigned arrival or departure procedures. To reduce noise pollution for the population living in the vicinity of the airport, descent angles are set so that aircraft do not approach the Earth's surface closely at a greater distance from the airport where urban development is denser. Similarly, the arrival and departure routes are chosen differently during evening hours to reduce noise pollution for residents. Furthermore, the analysis revealed variations in altitude preferences among different types of aircraft and operations (VFR, IFR). The distribution of these preferences may arise from two sources: firstly, from the aircraft operators who express a preference for flying along routes within the prescribed altitude limits, and secondly, from the ATSP, which affects the structure of the traffic in the way it manages it. These variations in altitude preferences reflect the diverse nature of traffic within the Ruzyně area CTR and provide valuable information for airspace planning and management.

While the impact on manned traffic is not the only parameter in Air Risk Assessment (ARA)), Prague has been shown to have significant potential for establishing U-space airspace, especially when considering traffic density as a crucial factor. The challenge may arise in areas with dense manned operations, as indicated by the results near the airport or at points published within the SID/STAR procedures. This could potentially pose a problem for the use of UAM in logistics in the future, as many logistics centers in large metropolises are located near airports.

Although the traffic density at CTR Ruzyne may vary throughout the day, it was not included in this analysis. It is expected that the distribution of UAM traffic throughout the day will be very similar to the current VFR traffic. Therefore, during hours when VFR traffic is high, UAM traffic is also expected to be high. In contrast, this will also be the case during low-traffic hours. On the other hand, air transport operates around the clock during the day, but its impact on UAM traffic is expected to be minor. Except in the vicinity of the airport.

The provision of data for the study appears to be sufficient for the selected airspace, driven by several factors. First, the radar data captured within the Ruzyně CTR, which, due to its controlled status and higher altitude, is considered adequate. It is also worth noting that it covers the majority of Prague's territory. Second, the airspace is a TMZ, meaning that there are data available for most manned traffic in that area. However, this is not a requirement for most (regional) airports. Lastly, the depiction does not just replicate IFR traffic, which can be evaluated through published SID/STAR routes, but also includes partial GA traffic and HEMS (Helicopter Emergency Medical Service) flights, regularly utilizing routes between hospitals, airports, and more.

The study for São Paulo [2] airport determined the locations of UAM operations based on identified arrival and departure clusters. A buffer was established around these routes and this buffer defined the areas for UAM operations. The study did not include observations of airspace that were only overflights. If overflights were also included in the methodology, the buffer would practically encompass the entire observed airspace.

Therefore, for this article, it was necessary to establish a threshold level of traffic density at which UAM operations can be conducted, even in areas with low levels of manned traffic.

Comparing the results of this analysis with previous ones is challenging, primarily due to the limited number and availability of studies conducted on similar topics. On the other hand, this analysis can serve as a starting point or a reference for future implementation of U-space airspace in the Ruzyně CTR (Control Zone), which can be expected due to its economic potential.

## 5. Conclusions

The aim of this article was to evaluate the suitability of establishing a U-space for UAM purposes in terms of traffic density in the CTR over the chosen metropolis. For this purpose, an air traffic analysis was conducted based on data from secondary radar transponders provided by the ANSP. Various types of operation were considered, including commercial aircraft, general aviation, security service flights, and air rescue services. The data provided were collected continuously over 24 h throughout the month. Therefore, all flights were recorded in these data. In total, 2306 unique call signs in the CTR of Ruzyně were analyzed for June 2022.

The analysis revealed that there are areas in the CTR over the chosen metropolis where the arrival and departure routes do not reach below 1000 ft. These areas include a significant part of the city center, where a high demand for transport services using UASs is expected in the future. This demand can be anticipated primarily due to efforts to reduce traffic congestion and emission pollution in the city center.

When processing data from the ASTERIX system, it was found that the data recording frequency averaged around 5 s. For some flights, there were delays of more than 5 s between transmissions. At the average speed of the aircraft during the approach phase, this delay could represent a distance of hundreds of meters between individual data points. This means that when plotting the data using grids, there could be situations where a faster moving aircraft passes through a grid without sending a transmission to the SSR. The data were considered to interpolate to fill in these gaps. However, during data analysis, it became apparent that some flights were not recorded in entirety, probably because the crew turned off the secondary radar transponder when leaving the TMZ and then turned it back on when entering the TMZ area. Therefore, the interpolation created data points that directly connected the entry and exit points of the CTR for some aircraft. The limitations of the data come primarily from their source, as the data were obtained from a single source, the Czech ANSP. However, it is worth noting that the data source is the largest provider of navigation services in the Czech Republic. The data includes all traffic in the Ruzyně CTR. Another limitation is the absence of call signs for some flights. Since call signs are a key distinguishing factor between individual flights, these data could not be included in the analysis. However, it is important to note that this represented approximately 3% of the records, which did not significantly affect the results of the analysis.

The analysis of air traffic in the Ruzyne CTR used data from June 2022. One limitation is that this period still saw some impact from the COVID-19 pandemic, resulting in slightly reduced air traffic. On the other hand, it is important to note that analyzing air traffic in the Ruzyně CTR requires current input data, and the analysis is needed now. Therefore, it is not feasible to wait for air traffic to fully return to prepandemic levels before conducting the analysis. Also, from the strategic point of view, it is not expected that the number of general aviation, security forces, or HEMS flights in the determined area shall increase drastically in the near future. For this reason, it is suitable to use the ANSP data provided to determine the operating volumes of the UAM.

Assessment of traffic density in the CTR above the selected metropolis is a necessary and valuable step toward the establishment of U-space for UAM purposes. This concept of transport within major urban areas is currently a widely discussed topic, especially in the context of sustainable mobility, as it offers the potential to transport people and cargo with

reduced environmental impact and carbon neutrality. Therefore, it can be expected that this mode of transportation will be introduced into daily life.

Traffic density is not the only important metric to consider when establishing U-space. Other parameters, such as ground risk, security, noise, and environmental impacts, should also be taken into account. Exploring these parameters offers a logical and meaningful next step of this research.

**Author Contributions:** Conceptualization, J.K. and A.K.; Methodology, J.K. and A.K.; Data Acquisition, A.K.; Data Analysis, M.Č., T.T., J.K. and A.K.; Proofreading, T.T., M.Č., J.K. and M.L.; Results Visualization, T.T. and M.Č.; Results Validation, J.K. and M.L.; Supervision, M.L. and J.K.; Funding Acquisition, J.K. All authors have read and agreed to the published version of the manuscript.

**Funding:** This research received no external funding.

**Institutional Review Board Statement:** Not applicable.

**Informed Consent Statement:** Not applicable.

**Data Availability Statement:** The data presented in this study are available on request from the corresponding author.

**Conflicts of Interest:** The authors declare no conflict of interest.

## Abbreviations

The following abbreviations are used in this manuscript:

| | |
|---|---|
| AAM | Advanced Air Mobility |
| ADS-B | Automatic Dependent Broadcast-Broadcast |
| AGL | Above Ground Level |
| ANSP | Air Navigation Service Provider |
| ATC | Air Traffic Control |
| ATSP | Air Traffic Service Provider |
| ASTERIX | All Purpose Structured Eurocontrol Surveillance Information Exchange |
| BVLOS | Beyond Visual Line of Sight |
| CTR | Control Zone |
| DAR | Dynamic Airspace Reconfiguration |
| DEM | Digital Elevation Model |
| GA | General Aviation |
| HEMS | Helicopter Emergency Medical Service |
| IFRs | Instrument Flight Rules |
| ISA | International Standard Atmosphere |
| PSR | Primary Surveillance Radar |
| SSR | Secondary Surveillance Radar |
| TMA | Terminal Control Area |
| TMZ | Transponder Mandatory Zone |
| UAM | Urban Air Mobility |
| UASs | Unmanned Aircraft Systems |
| UTM | Unmanned Traffic Management |

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
