# Peer review of "Evaluating U-Space for UAM in Dense Controlled Airspace"

_drones, doi:10.3390/drones7120684_

Round 1

Reviewer 1 Report

Comments and Suggestions for Authors

In future, the air space should be controlled for various vehicles by using ATM and UTM(UATM).  For the UAM, there must be a comprehensive things for setting up the operational corridors by considering available vehicle's performance and flight capabilities, the social and environmental resitrictions, and the political situations. How to integrate the ATM and UTM for the control of future airspace is a huge task because the AAM or UAM is an emerging technology and the regulation has not established.

However, current aspects for evaluating the U-space in the dense controlled airspace can give one of those considerations for deciding the corridors. 

Present study consider the impact of the introduction of UAS into the controlled dense space by using the the analysis of ANSP data under 1000ft(AGL) which is the minimum safe altitude defined by ICAO. Literature survey was accomplished by including recent publications. The results in the present study can be a more specific case study limited to Prague, Czech Republic, but its population is more than 2 millions. Thus, present paper can be of great help to the related researchers.

I have several suggestions. For the UAM, how to integrate the ATM and UTM for the controlled airspace is a huge task because the AAM or UAM is an emerging technology and the regulation on the airspace is under establishment. I am wondering if the authors can split the data at least two altitudes (500ft and 1000ft) for future applications. 

The UAM will operate when the traffic congestion is high. Thus I am wondering if the authors can show the data by the time of the day.

Author Response

Thank you for your review of the article. Thank you for your suggestions for including two different altitudes (500 ft and 1000 ft). We included additional results for 500 ft AGL. (Figure 4-7)

Discussion of the inclusion of the data by the time of the day has been added to the Discussion Section (Line 359-360).

Reviewer 2 Report

Comments and Suggestions for Authors

The authors address a problem of considerable importance for the future of Urban Air Mobility. In particular, they analyze the airspace in an area with dense controlled airspace to evaluate the areas with lower risk and suitable for UAM.

However, I have some comments and suggestions to improve the paper:

- The introduction mainly focuses on the U-Space description and analysis. However, I suggest also including a literature review on studies about air risk assessment with general aviation and UASs

- The analysis of traffic data is performed for flights below 1000ft. However, it can be interesting to analyze the flight altitude distribution, since the traffic can change a lot considering different altitude levels

- In my point of view, a more advanced analysis should be performed, e.g. statistical analysis.

- It can be interesting to evaluate also the Mid-Air Collision Rate in the area of interest.

- line 250, Table ???

- In my point of view, also the ground risk affected by mid-air collisions should be considered for the definition of suitable airspace for UAM. In fact, flights over high population density areas can expose people at risk.

Comments on the Quality of English Language

The paper is generally well-written. However, there are some paragraphs that can be improved from a language point of view

Author Response

Thank you for your review of the article. Thank you also for your suggestions to include in the article. 

The article's introduction was extended and improved to include an extended literature review regarding air risk assessments conducted in the past. Also, studies about air risk probability in manned aviation were added. 

Thank you for your suggestion to include more altitudes below 1000 ft. Additional figures and results were added regarding heights 0 to 500 ft. 

Changes:
Literature review regarding air risk assessments has been added to the introduction. (Line 118-126)

Results for altitude of 500 ft AGL has been added. (Figures 4-7 + Table 2)

Table reference has been added to the Line 296 (previously Line 250).

Ground risk affected by mid-air collision has been discussed in the introduction. (Line 110-116)

Reviewer 3 Report

Comments and Suggestions for Authors

This study explores a fine-grained suitability analysis of establishing U-space for UAM operation in the City of Prague. The authors leverage air traffic data, DEM data, and other supportive materials to assess the suitability of various types of UAM operations. The highlight of this paper is a thorough comparison of various air traffic data sources.

However, the research can be improved in a number of ways before proceeding to publication.

1. The study should include a thorough literature review that places the study in the context of UAM/AAM/drone/UTM management research. Please consider the following flagship studies and other publications in the field.

R. Rumba and A. Nikitenko, "The wild west of drones: a review on autonomous- UAV traffic-management," 2020 International Conference on Unmanned Aircraft Systems (ICUAS), Athens, Greece, 2020, pp. 1317-1322, doi: 10.1109/ICUAS48674.2020.9214031.

Bosson, C., & Lauderdale, T. A. (2018). Simulation evaluations of an autonomous urban air mobility network management and separation service. 2018 Aviation Technology, Integration, and Operations Conference. https://doi.org/10.2514/6.2018-3365 

Straubinger, A., Rothfeld, R., Shamiyeh, M., Büchter, K., Kaiser, J., & Plötner, K. (2020). An overview of current research and developments in urban air mobility – Setting the scene for UAM introduction. Journal of Air Transport Management, 87, 101852. https://doi.org/10.1016/j.jairtraman.2020.101852 

Nelson, J. R., & Gorichanaz, T. (2019). Trust as an ethical value in emerging technology governance: The case of drone regulation. Technology in Society, 59, 101131. https://doi.org/10.1016/j.techsoc.2019.04.007 

Li, X., & Kim, J. H. (2022). Managing disruptive technologies: Exploring the patterns of local drone policy adoption in California. Cities, 126, 103736. https://doi.org/10.1016/j.cities.2022.103736

2. Please justify the use of current air space traffic data to analyze future UAM operation. What are the pros, cons, and limitations of this approach?

3. The study should rewrite the method section to clearly describe the data sources, scenario design, and models. Please consider use graphs and tables instead of text only.

4. The English writing in the paper needs further polishing.

Comments on the Quality of English Language

The English writing in the paper needs further polishing.

Author Response

Thank you for your review of the article. Thank you for suggesting various studies to include in the article. They have been read through and included in the article, in the introduction part.

Then the use of the current airspace data has been justified. The pros, cons and limitations were described.

The methodology sections was improved to show more clear approach to the data sources and also data processing. Multiple figures and tables were added to make understanding of the text easier for the reader.

Furthermore, English writing was once more checked and improved.

Changes:
Literature review has been extended using the articles provided by the reviewer. (Line 26-27 + Line 33-36)

The use of current airspace data has been justified. (Line 257-272 + Line 415-423).

Methodology sections has been rewritten. It was decided to use subsection to make the text easier to read and understand for the readers. Also graphs and tables were added for the same reason. (Line 166 + Line 201 + Line 219 + Line 192).

The English writing has been once again checked and polished. 

Round 2

Reviewer 2 Report

Comments and Suggestions for Authors

The authors revised the paper according to the comments made. The state of the art has been improved, they have also considered lower flight altitudes in the results (e.g. 500 ft) and other minor improvements.

They did not extend the analysis of the results, which could have been done through a probabilistic analysis to have a more scientific soundness.

In any case, this may require more time and work to be done in future work.

Comments on the Quality of English Language

Some grammatical errors and typos are present.

Author Response

Thank you for the revision of the article.

Changes:
The more complex probabilistic analysis can be conducted in the future. This would need more time than was provided for the revision of the article. Moreover, it would be needed to have more data on the higher altitudes for it to analyse the possibilities of U-space implementation in higher altitudes. The goal of this article was to conduct a study regarding U-space in Very Low Level Airspace, where the division to 500 ft and 1000 ft is sufficient. 

The English writing has been improved.

Reviewer 3 Report

Comments and Suggestions for Authors

Please double-check the format and content of the reference list. For instance, reference [2] and reference [17] are the same article. Regarding the literature pertaining UAM infrastructure planning, there are a number of pioneering articles published on Drones journal which can be potentially added to the literature review section.

Comments on the Quality of English Language

The writing has been moderately improved.

Author Response

Thank you for the revision of the article.

Changes:
The reference list has been checked. The references has been corrected.

Additional literature review has been added to the article. 

The English writing has been widely improved.